# Unusual Surgical Repair of Bronchoesophageal Fistula Following Esophagectomy

**DOI:** 10.3390/diagnostics14212432

**Published:** 2024-10-30

**Authors:** Predrag Sabljak, Ognjan Skrobic, Aleksandar Simic, Keramatollah Ebrahimi, Dejan Velickovic, Vladimir Sljukic, Nenad Ivanovic, Milica Mitrovic, Jelena Kovac

**Affiliations:** 1Department of Stomach and Esophageal Surgery, Clinic for Digestive Surgery, University Clinical Centre of Serbia, Koste Todorovica Street No. 6, 11000 Belgrade, Serbia; predragsabljak63@gmail.com (P.S.); apsimic65@gmail.com (A.S.); keramatollahe@gmail.com (K.E.); velickovicdejan@gmail.com (D.V.); vlada.sljukic@gmail.com (V.S.); nekic85@gmail.com (N.I.); 2Department for Surgery, Faculty of Medicine, University of Belgrade, Dr Subotica No. 8, 11000 Belgrade, Serbia; 3Center for Radiology and Magnetic Resonance Imaging, University Clinical Centre of Serbia, Pasterova No. 2, 11000 Belgrade, Serbia; dr_milica@yahoo.com (M.M.); jelenadjokickovac@gmail.com (J.K.); 4Department for Radiology, Faculty of Medicine, University of Belgrade, Dr Subotica No. 8, 11000 Belgrade, Serbia

**Keywords:** esophagectomy, bronchoesophageal fistula, airway–gastric fistulas

## Abstract

Radical esophagectomy remains the only potentially curative option in the treatment of esophageal cancer. However, this procedure is burdened with high morbidity and mortality rates, even in high-volume centers. A tracheo- or bronchoesophageal fistula (TBF) is rare but is one of the most difficult life-threatening complications following an esophagectomy for cancer treatment. Several classifications have been proposed regarding the localization of a TBF, its etiology, and the timing of its occurrence; hence, no classification is universally accepted. However, one of the most common etiological explanations for the formation of a TBF is a prior esophagogastric anastomotic leak. Treatment options include a conservative approach, which usually combines several endoscopic methods. Surgical treatment is directed towards fistula closure with direct suturing or, more often, the usage of pediculated flaps. Here, we present a patient with late TBF following a minimally invasive esophagectomy, which was surgically solved in an atypical way. We believe that this type of repair may be useful in patients in whom pedunculated flaps are not an option.

Esophageal cancer represents the sixth most common malignant disease, with a rapid development and poor survival rates. Radical surgery, including esophagectomy with reconstruction, remains the mainstay as it is the only potential curative treatment. Nowadays, neoadjuvant chemotherapy or chemoradiotherapy is usually the standard of care for tumors beyond preoperative T2N0 stage.

A 64-year-old female was diagnosed with mediodistal squamocellular cancer of the esophagus. Chest and abdomen multi-detector computed tomography (MDCT) was performed, revealing enlarged lymph nodes in the distal mediastinum and a 45 mm long tumor at local stage T3. Preoperatively, neoadjuvant chemoradiotherapy (CRT) was administered. The protocol delivered radiation in 25 fractions to a total dose of 42.1 Gy, along with concomitant 5FU/CDDP (5-fluoracil/cisplatinum) chemotherapy (CT). MDCT showed signs of partial tumor regression, and we decided to move forward with a radical esophagectomy. The minimally invasive Ivor Lewis procedure was performed, with a two-field standard lymphadenectomy and an esophagogastric anastomosis at the level of the tracheal carina. The postoperative course was uneventful; the patient was discharged 8 days after surgery in good condition, and she was capable of normal peroral food intake. The histopathological finding corresponded to stage T3N0 with a partial response to chemoradiotherapy.

One month after the surgery, the patient underwent adjuvant CT (5FU/CDDP); however, due to severe neutropenia as a side effect, only one cycle was administered.

Then, 45 days after surgery, the patient was admitted to a regional hospital due to severe dyspnea, coughing, and vomiting, as well as clinical signs of pneumonia. Laboratory findings showed elevated levels of C-reactive protein (CRP) and white blood cell count (WBC). She was then transferred to our department for further evaluation. Upper gastrointestinal (GI) endoscopy was performed, showing a fistula opening to the airway at the level of the esophagogastric anastomosis (Figure 1). At this point, the presence of a tracheobronchial fistula (TBF) was suspected. A chest X-ray of the thorax showed signs of pneumonia.

Subsequently, an esophagogram showed the leakage of the contrast medium into the tracheobronchial lumen, demonstrating the presence of a TBF (Figure 2). This was confirmed via MDCT and bronchoscopy. An NG tube was placed for feeding and decompression of the gastric conduit. Antibiotics meropenem (1 g, 8 h) and vancomycin (1 g, 12 h) were administered with the addition of parenteral nutrition.

After the resolution of pneumonia, we evaluated the patient. Due to the size and localization of the fistula, no conservative management was possible, and we decided to move forward with the surgical repair of the fistula. Surgery was performed via a right posterolateral thoracotomy, and the gastric tube was disconnected from the esophagus, exposing a defect on the left main bronchus that was 18 mm in diameter. At this time, the patient was very thin; thus, there was no possibility of using the intercostal muscle flap to cover the defect. We decided on the only possible solution: using the posterior wall of the esophagus, proximal to the anastomosis. The esophagus was transected in such a manner that a posterior mucosal flap was created, which was 4 cm long and received a blood supply coming from the posterior esophageal wall (Figure 3). The flap was well vascularized and long enough. We covered the defect with the flap using absorbable 4.0 sutures. An air leak test was performed, showing no leak from the repaired fistula. The procedure was finished with a left cervicotomy and esophagostomy, as well as a median laparotomy and gastrostomy, which were performed on the partially resected gastric tube previously repositioned into the abdomen.

Postoperatively, the patient recovered well and was discharged two weeks after the surgery. At the time of the one-month follow-up, the patient was relieved of her prior respiratory symptoms. Food intake was managed via nutritive gastrostomy.

With the hope of giving this patient a better quality of life, she was admitted three months later for reconstructive surgery. Prior to surgery, bronchoscopy was performed, showing a well-vascularized flap covering the defect on the left main bronchus (Figure 4A). Retrosternal colon interposition was performed using the transversosplenic segment of the colon as a graft, with blood supply provided by the left colic artery. The postoperative course was uneventful. On the 7th postoperative day, a contrast barium study was performed, showing normal passage through the colon graft (Figure 4B). Per oral intake was then initiated.

The patient has been under regular surveillance since. One and a half years after the esophagectomy, she is disease-free. She reports having a good quality of life and swallowing function.

Esophagectomy with gastric pull-up reconstruction remains the only potentially curative treatment method for esophageal cancer. Despite advances in surgical techniques, this procedure is burdened with high morbidity rates, even in high-volume centers [1]. Tracheo- and bronchoesophageal fistulas (TBFs) following esophagectomy are one of the most fearsome complications of esophagectomy. TBFs are almost always accompanied by severe pneumonia, mediastinitis, and sepsis. Solving these dreadful complications remains complex and lacks specific treatment guidelines; however, some recommendations have been proposed [2].

The development of tracheo- or bronchoesophageal fistulas after esophagectomy is one of the most severe complications and is burdened with high mortality rates. These fistulas are usually the result of a prior leak from esophagogastric anastomosis. The etiology of a TBF after esophagectomy is probably multifactorial [3,4]. Anastomotic leakage seems to be the leading risk factor for leak development; however, the incidence of anastomotic leakages after esophagectomy is much higher than TBF, meaning that there is probably more than one mechanism involved. Several studies have postulated that extensive dissection alongside the tracheal carina and main bronchi may facilitate ischemic lesion formation in the airway [5]. Furthermore, a lung ventilation cuff of the double-lumen tube may aggravate partial ischemia of the membranous left main bronchus wall and facilitate thermal injury during surgical dissection.

If a TBF develops at the same time as an anastomotic leak, it is classified as synchronous; if it develops some period after the anastomotic leak, it is classified as a metachronous fistula [3]. This classification was proposed by Lambertz et al., who also found the left main bronchus to be the most common opening for a TBF, followed by the trachea and right main bronchus. The authors also considered fistulas to be direct if the airway opening was on the trachea and indirect if the orifice of the TBF was on the left main bronchus [3]. In our patient, we have a case of a metachronous indirect fistula, which developed a month and a half after esophagectomy. It is our assumption that the patient developed a subclinical anastomotic leak, progressing to ulceration and, subsequently, invasion of the left main bronchus.

A second classification for tracheobronchial necrosis following esophagectomy was proposed by Sakai et al. [6]. The authors suggested type I and II fistulas to be those characterized by tracheal necrosis of either the proximal or distal parts of the trachea after tracheostomy. Type III was further characterized by necrosis of the lower trachea without tracheostomy, and type IV was considered to be bronchial necrosis. Our patient would be classified as having a type IV fistula: one of the most difficult to solve. Different classifications of airway–gastric fistulas (AGFs) after esophagectomy were proposed by Wang and coauthors [7]. Based on a large retrospective study, these authors classified AGFs according to the anatomical height of the orifice on the gastric tube and airway. Type I was considered if the orifice of an AGF was higher in the gastric tube than in the airway, and type II was considered in cases where both the gastric and airway orifices were in a horizontal plane. The authors proposed that a type I AGF may be treated conservatively and that type II AGFs require surgical treatment. Therefore, our patient would be classified as type II, and our treatment strategy, including a surgical approach, is in concordance with the suggestions of that study.

There are no standardized treatment guidelines for TBFs following esophagectomy. Initial management includes the treatment of pneumonia, and it often includes prolonged intubation and mechanical ventilation. Conservative management is usually possible in cases with smaller leak openings. Various methods have been described, including the usage of digestive and airway stents, fibrin glue and its variations, as well as endoscopic clips [8,9,10]. Palmes et al. reported a series of 15 patients with TBFs following esophagectomy. The authors concluded that conservative treatments can often fail in more than half of patients, resulting in sepsis and poor outcomes [11]. Surgical solutions are recommended when patients have a stable condition, warranting better outcomes, as a one-stage procedure.

Surgical treatment is demanding and still burdened with high mortality rates. Often, the gastric conduit needs to be sacrificed in order to achieve good exposition of the fistula. Direct suture of the airway opening is feasible in smaller openings, with different types of pediculated flaps proposed for covering a defect. Commonly, the intercostal muscle flap or pedunculated pericardial flaps are used [12,13]. In this specific case, we had a very thin patient, and it was our assumption that the intercostal flap would not be sufficient to cover the defect. Therefore, we used the esophageal flap proximal from the opening of the left main bronchus, which had sufficient length and vascularization and managed to cover the defect on the left main bronchus successfully.

Potential risk factors for the development of TBFs after esophagectomy were evaluated from the single study that we were able to identify in the literature. There was no specific risk factor, including the type of surgical procedure or neoadjuvant chemotherapy or chemoradiotherapy prior to esophagectomy, which could be related to the occurrence of TBFs [14].

To conclude, a TBF remains a difficult clinical problem, burdened with high mortality rates. An individual approach for every patient, as well as a combination of conservative endoscopic and surgical management, needs to be employed in order to achieve the best possible outcomes.

## Figures and Tables

**Figure 1 diagnostics-14-02432-f001:**
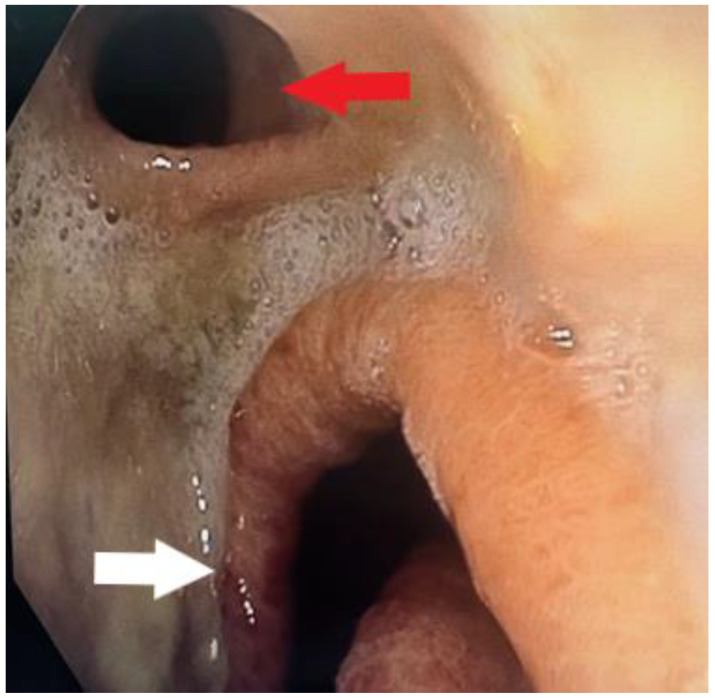
Upper GI endoscopy revealing a fistula orifice (red arrow) at the level of the esophagogastric anastomosis. A gastric tube was placed below (white arrow). At this point, it was obvious that endoscopic treatment of the fistula was not possible. The presence of fibrin around the fistula was indicative of a previous ulcer at the level of the esophagogastric anastomosis and a possible leak that had been clinically undetected.

**Figure 2 diagnostics-14-02432-f002:**
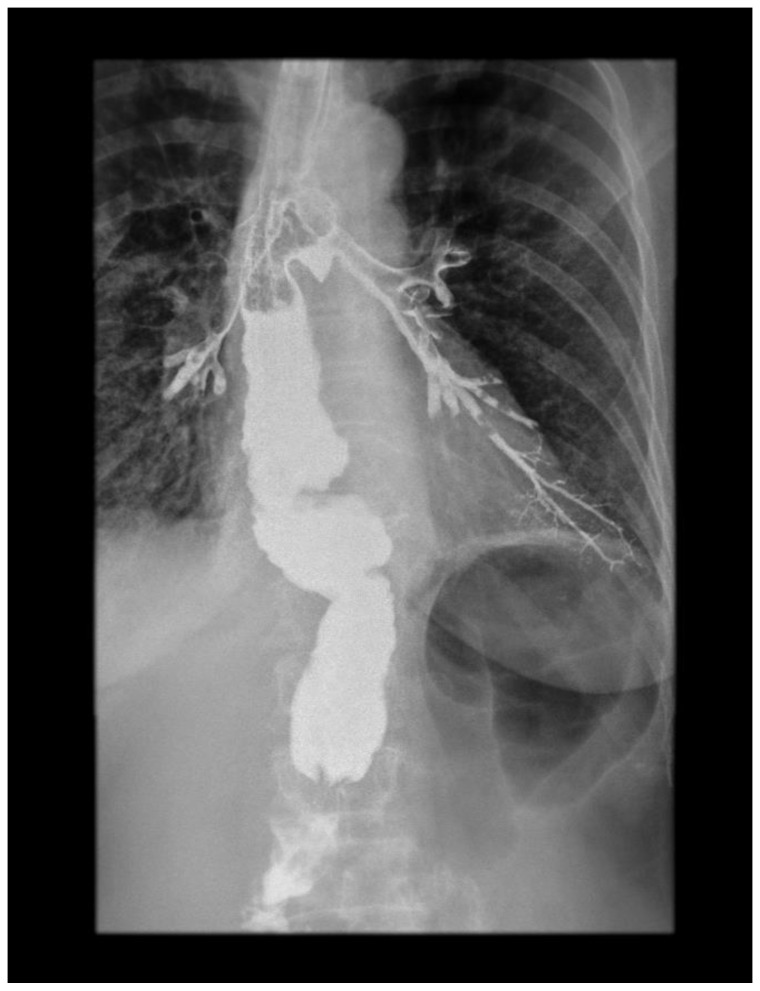
Contrast X-ray study showing the gastrobronchial leak. The diagnostic modalities include barium swallow, CT imaging, bronchoscopy, and upper gastrointestinal endoscopy. Identifying TEF can be challenging. Esophagoscopy can often miss small or discrete fistulas, even with fluoroscopy. Contrast radiography is a very useful method in terms of establishing this diagnosis. It is actually the confirmatory test for TEF. The appearance of barium in the lumen of the tracheobronchial tree is a direct sign of the fistulization of these structures within the esophagus. This picture shows a clear esophagobronchial leak, which provoked an irritating cough during the patient’s examination.

**Figure 3 diagnostics-14-02432-f003:**
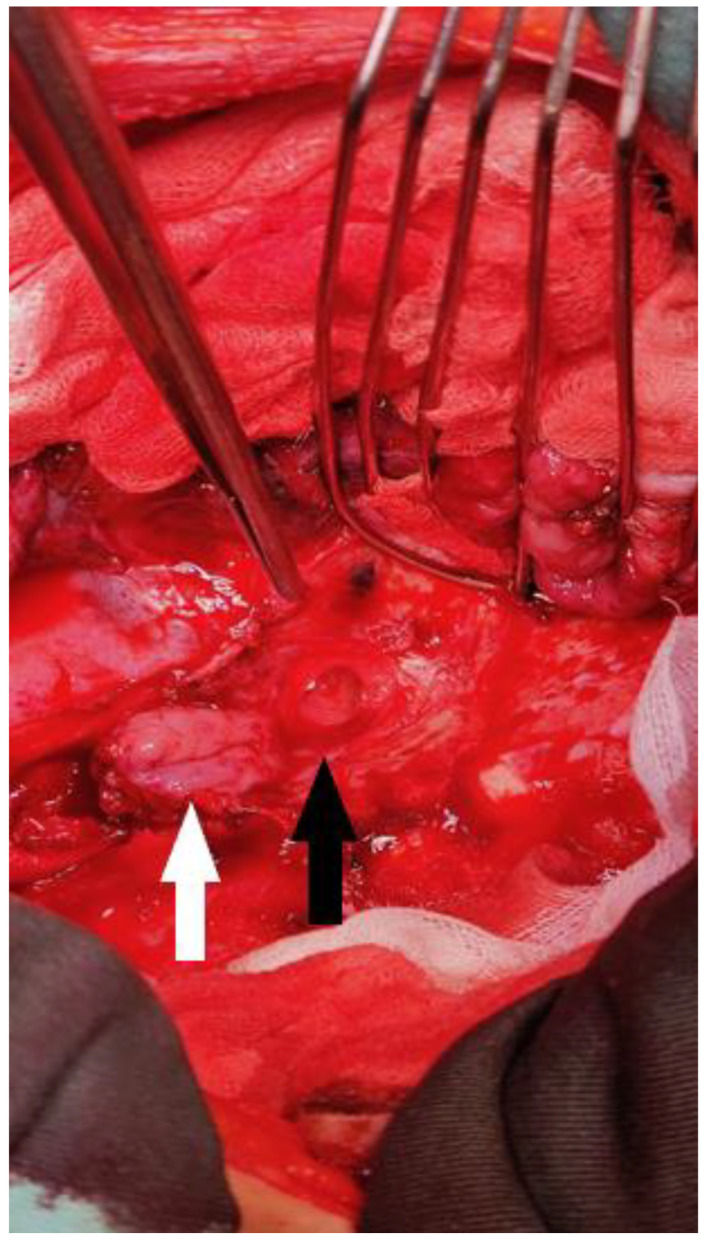
Intraoperative image showing a defect on the left main bronchus (black arrow) and the esophageal flap prepared for suturing (white arrow). This image was taken after the gastric tube had been disconnected and the esophageal flap prepared. The surrounding tissue was fibrotic. The esophageal flap was well vascularized and had sufficient length.

**Figure 4 diagnostics-14-02432-f004:**
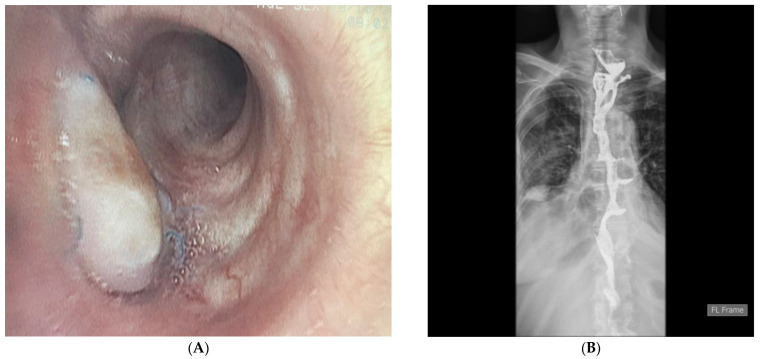
Bronchoscopy image of the flap covering the defect on the left main bronchus one month after surgery. The flap is well vascularized. The patient did not experience any respiratory symptoms after the surgery (**A**). Contrast X-ray study 7 days after colon interposition. The esophagocolonic anastomosis is wide; the colon graft is positioned in a linear fashion. There is no contrast leak, and the evacuation of the contrast into the stomach remnant is smooth (**B**).

## Data Availability

The datasets used and analyzed in this paper are available from the corresponding author upon reasonable request.

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
