# Peer review of "Unusual Surgical Repair of Bronchoesophageal Fistula Following Esophagectomy"

_diagnostics, 2024, doi:10.3390/diagnostics14212432_

Round 1
Reviewer 1 Report
Comments and Suggestions for Authors
Very interesting case report on a challenging clinical situation. Overall well described and well illustrated case.
Some suggestions:
- Could your report the histological staging of the tumor after the esophagectomy?
- Is adjuvant CT your standard protocol after neoadjuvant chemoradiotherapy and esophagectomy? Please comment on this treatment approach.
- Is it correct that an NG tube was placed for feeding and patient also received parenteral nutrition? If so, why were both administered. And please explain why a surgical feeding jejunostomy was not placed at the time of esophagectomy.
The quality of the English language is poor. Many grammatical errors. This is very distracting when reading the article.
Author Response
Dear,
We would like to express our sincere appreciation for your valuable comments on our paper “Unusual surgical repair of bronchoesophageal fistula following esophagectomy”. We have given the comments serious consideration and corrected the manuscript according to the suggestions.
We hope that the revised manuscript will meet your expectations and we are willing to consider all further revisions. The revisions have been approved by all authors and the revised manuscript is attached. Thank you for your interest in our manuscript!
Comments and answers:
Reviewer 1
- Could your report the histological staging of the tumor after the esophagectomy?
Authors: Thank you for this comment, we have supplemented the introductory part with the requested data. Ln 46-47.
- 2. Is adjuvant CT your standard protocol after neoadjuvant chemoradiotherapy and esophagectomy? Please comment on this treatment approach.
Authors: Thank you for pointing this out to us. In fact, it is not our standard protocol. In this particular case, the patient was sent for adjuvant chemotherapy by medical oncologists from the regional hospital without our specific knowledge.
- 3. Is it correct that an NG tube was placed for feeding and patient also received parenteral nutrition? If so, why were both administered. And please explain why a surgical feeding jejunostomy was not placed at the time of esophagectomy.
Authors: NG tube was placed for feeding and decompression of the gastric conduit. This was corrected in the manuscript (ln 66-67) We commonly use nasojejunal tube for this puropose, however patient had poor tolerance of the tube, so we had to add parenteral nutrition as well, to prepare her adequatly for surgery. Feeding jejunostomy is not our standard approach, as we have a very low incidence of esophagogastric anastomotic leak.
- 4. The quality of the English language is poor. Many grammatical errors. This is very distracting when reading the article.
Authors: Bearing in mind your comment about the poor quality of the English language, we decided to use MDPI language correction service. Thank you for your suggestion.
This manuscript was corrected and explained in more details in view of the requirements of the reviewers. Thank you for the constructive advices and comments. We hope that the revised manuscript will receive an affirmative answer.
Reviewer 2 Report
Comments and Suggestions for Authors
Title: Unusual surgical repair of bronchoesophageal fistula following esophagectomy
Thank you for submitting your manuscript. After an in-depth review, I believe that your work has the potential to be published with considering the below major revisions:
· The Abstract is not representative of the importance and necessity of the paper; revise it, please!
· At least 2-3 paragraphs should be written regarding the disease definition and diagnosis, importance, and treatment options (As you discussed some of these general aspects in the discussion part). Then you can continue with the case presentation. After that the discussion should cover the importance of your case and its related aspects.
· Line 59: add the details about course of antibiotic therapy for pneumonia.
· The manuscript should be revised by a native English speaker. Instead of using complex words, pay more attention to the clarity of the text and spelling.
o Line 26: …present a patient….
o Line 38: lymhadenectomy àlymphadenectomy?
o Line 54: no possibleà not possible
o Line 74-75: has grammatical errors and need to be revised.
o Past tense should be used regarding patient’s status!
Comments on the Quality of English LanguageThe manuscript should be revised by a native English speaker. Instead of using complex words, pay more attention to the clarity of the text and spelling.
o Line 26: …present a patient….
o Line 38: lymhadenectomy àlymphadenectomy?
o Line 54: no possibleà not possible
o Line 74-75: has grammatical errors and need to be revised.
o Past tense should be used regarding patient’s status!
Author Response
Dear,
We would like to express our sincere appreciation for your valuable comments on our paper “Unusual surgical repair of bronchoesophageal fistula following esophagectomy”. We have given the comments serious consideration and corrected the manuscript according to the suggestions.
We hope that the revised manuscript will meet your expectations and we are willing to consider all further revisions. The revisions have been approved by all authors and the revised manuscript is attached. Thank you for your interest in our manuscript!
Comments and answers:
Thank you for having the time to carefully read our manuscript. Your comments will improve the manuscript significantly. Please find response to your comments in the text bellow.
Reviewer 2
- 1. The Abstract is not representative of the importance and necessity of the paper; revise it, please!
Authors: We completed the abstract and structured it differently.
- At least 2-3 paragraphs should be written regarding the disease definition and diagnosis, importance, and treatment options (As you discussed some of these general aspects in the discussion part). Then you can continue with the case presentation. After that the discussion should cover the importance of your case and its related aspects.
Authors: We added 3 sentences in the introduction part regarding the disease definition and diagnosis, as well as the treatment options. (Added text ln 30-34)
- Line 59: add the details about course of antibiotic therapy for pneumonia.
Authors: We specified the antibiotic therapy and added it to the manuscript, Ln 67.
- The manuscript should be revised by a native English speaker. Instead of using complex words, pay more attention to the clarity of the text and spelling.
Line 26: …present a patient….
Line 38: lymhadenectomy àlymphadenectomy?
Line 54: no possibleà not possible
Line 74-75: has grammatical errors and need to be revised.
Past tense should be used regarding patient’s status!
Authors: Thank you for the suggestion that we should correct the quality of the English language. We opted for the MDPI language correction service.
This manuscript was corrected and explained in more details in view of the requirements of the reviewers. Thank you for the constructive advice and comments. We hope that the revised manuscript will receive an affirmative answer.
Round 2
Reviewer 2 Report
Comments and Suggestions for Authors
It is accepted.